# Rat donor lung quality deteriorates more after fast than slow brain death induction

Judith E. van Zanden[1]*, Rolando A. Rebolledo[1,2], Dane Hoeksma[1], Jeske M. Bubberman[1], Johannes G. Burgerhof[3], Annette Breedijk[4], Benito A. Yard[4], Michiel E. Erasmus[5], Henri G. D. Leuvenink[1], Maximilia C. Hottenrott[1,6]

**1** Department of Surgery, University of Groningen, University Medical Center Groningen, Groningen, The Netherlands, **2** Institute for Medical and Biological Engineering, Schools of Engineering, Biological Sciences and Medicine, Pontificia Universidad Católica de Chile, Santiago, Chile, **3** Department of Epidemiology, University of Groningen, University Medical Center Groningen, Groningen, The Netherlands, **4** Department of Internal Medicine, V. Clinic, University Medical Center Mannheim, Mannheim, Germany, **5** Department of Cardiothoracic Surgery, University of Groningen, University Medical Center Groningen, Groningen, The Netherlands, **6** Department of Surgery, University of Regensburg, Regensburg, Germany

* j.e.van.zanden@umcg.nl

**Data Availability Statement:** All relevant data are within the manuscript and its Supporting Information files.

## Abstract

Donor brain death (BD) is initiated by an increase in intracranial pressure (ICP), which subsequently damages the donor lung. In this study, we investigated whether the speed of ICP increase affects quality of donor lungs, in a rat model for fast *versus* slow BD induction. Rats were assigned to 3 groups: 1) control, 2) fast BD induction (ICP increase over 1 min) or 3) slow BD induction (ICP increase over 30 min). BD was induced by epidural inflation of a balloon catheter. Brain-dead rats were sacrificed after 0.5 hours, 1 hour, 2 hours and 4 hours to study time-dependent changes. Hemodynamic stability, histological lung injury and inflammatory status were investigated. We found that fast BD induction compromised hemodynamic stability of rats more than slow BD induction, reflected by higher mean arterial pressures during the BD induction period and an increased need for hemodynamic support during the BD stabilization phase. Furthermore, fast BD induction increased histological lung injury scores and gene expression levels of TNF-α and MCP-1 at 0.5 hours after induction. Yet after donor stabilization, inflammatory status was comparable between the two BD models. This study demonstrates fast BD induction deteriorates quality of donor lungs more on a histological level than slow BD induction.

## Introduction

Lung transplantations are generally performed with lungs derived from brain-dead donors, who suffered from extensive central nervous system injury secondary to trauma, hemorrhage or infarction [1, 2]. An interesting observation in lung donation is the lower procurement rate compared to other solid organs, with an acceptance rate of 56% for lungs in contrast to 76% for livers and 82% for kidneys [3, 4]. Besides multiple risk factors for donor lung injury such as mechanical ventilation, aspiration and infection, the process of brain death (BD) is described to induce lung damage [5–7].

**Funding:** The authors received no specific funding for this work.

**Competing interests:** The authors have declared that no competing interests exist.

BD is initiated by an increase in intracranial pressure (ICP), which leads to ischemia of the brain and brainstem. Subsequently, a massive release of catecholamines in the blood occurs, also known as the 'autonomic storm'. This phase is accompanied by a severe increase in systemic vascular resistance (SVR) [8–10]. The sudden change in SVR results in pooling of a large proportion of blood in the cardio-pulmonary vasculature. Shortly after this autonomic storm, SVR decreases and the aortic blood flow normalizes or even results in hypotension, of which the latter is seen in most subjects [11]. Along with hemodynamic changes, BD induces a pro-inflammatory environment. Cytokine formation and complement activation lead to a systemic inflammatory response (SIRS), which further damages peripheral organs [12–14]. As for the lungs, both early hemodynamic changes as well as the pro-inflammatory immune response are described to cause pulmonary edema and capillary leakage [15–17]. Eventually, BD-related changes contribute to inferior outcomes after transplantation [18].

The observed BD-related hemodynamic changes differ between various causes of BD, all of which correspond with different speeds of ICP increase. Clinically, traumatic brain injury is the most common cause of a fast ICP increase, while cerebrovascular events such as hemorrhagic stroke usually refer to a slower increase in ICP [19, 20]. Whether the speed of intracranial pressure increase contributes to the degree of donor lung damage, remains to be clarified. The aim of this study is to elucidate whether the speed of ICP increase affects quality of donor lungs from brain-dead donors, investigated in a rat model for fast *versus* slow BD induction. We showed that fast BD induction deteriorates quality of donor lungs more on a histological level than slow BD induction.

## Materials and methods

### Rats

Male Fischer (F344) rats weighing 270–300 g were obtained from Harlan Netherlands B.V. (Melderslo, the Netherlands). Before start of the experiments, rats were acclimatized for 1 week. Rats were housed under clean conventional conditions, in groups of 3 to enable social interactions. Standard rat chow was *ad libitum* available. Rats received humane care in compliance with the Principles of Laboratory Animal Care (NIH Publication No.86-23, revised 1996) and the Dutch Law on Experimental Animal Care. The experimental protocol was approved by the Institutional Animal Care and Use Committee–Rijksuniversiteit Groningen (IACU-C-RUG), approval No. 6645. All operations were performed under general anesthesia to minimize animal suffering.

### Experimental groups

Prior to the experiment, power analyses were performed to define the minimal amount of rats needed per experimental group. IL-6 was defined as the primary endpoint, since previous experiments in our laboratory showed that IL-6 is increased a 100-fold in brain-dead rats compared to sham-operated rats. With an absolute difference of 50%, a variability of 0.3 and power of 0.9, 8 rats per group were required. Rats were randomly assigned to 3 donor groups, (Fig 1): 1) control (immediate sacrifice, n = 8), 2) fast BD induction (ICP increase over 1 min, n = 32) or 3) slow BD induction (ICP increase over 30 min, n = 32). Group 2 and 3 served as BD models and brain-dead rats were sacrificed at 4 different time points: 0.5 hours, 1 hour, 2 hours and 4 hours after confirmation of BD. Humane endpoints were defined as: >15% loss of bodyweight prior to the experiment, e.g. due to stress and behavioral changes such as reduced exploratory activity. During preparatory steps for brain death induction, depth of anesthesia was confirmed by absence of muscle movement after toe pinch assessments. After induction, BD was confirmed by absence of corneal reflexes. No rats required euthanasia at humane

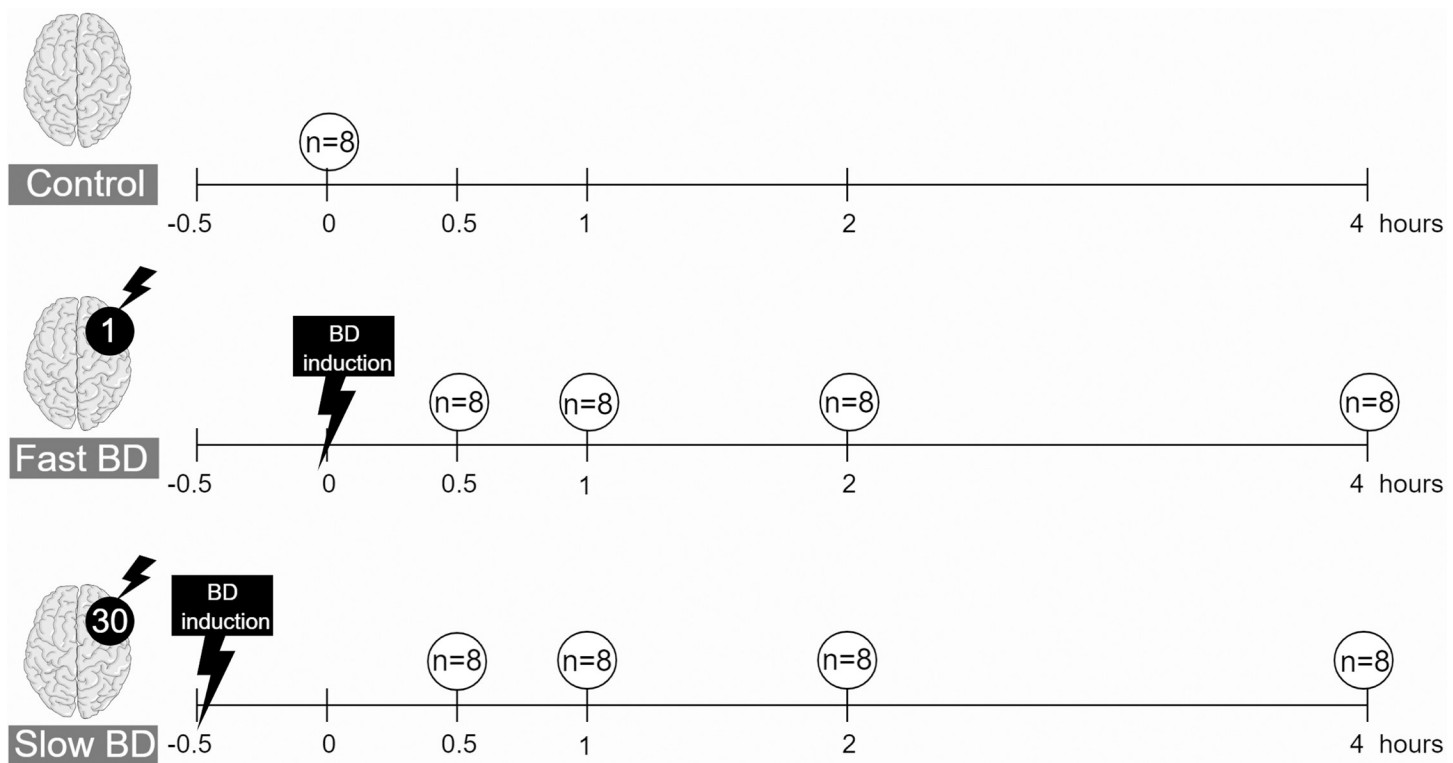

**Fig 1. Experimental outline of the study.** Rats were randomly assigned to 3 donor groups: 1) control (immediate sacrifice), 2) fast brain death (BD) induction (intracranial pressure (ICP) increase over 1 min) or 3) slow BD induction (ICP increase over 30 min). Rats subjected to BD were sacrificed at 4 different time points: 0.5 hours, 1 hour, 2 hours and 4 hours after BD induction. Figure created in the Mind the Graph platform: www.mindthegraph.com.

endpoints. Six rats were lost in the fast BD group due to inability of ventilation, as a consequence of fulminant lung edema immediately after BD induction. The cause of death was considered a direct result of fast BD induction and lost rats were replaced. No other rats were lost prior to our experimental endpoint.

## Rat brain death model

The BD procedure was based on previously described models [21, 22]. Isoflurane was selected for anesthesia and analgesia of rats, given its combination of hypnotic, analgesic and muscle relaxant properties [23]. Anesthesia was induced with a mixture of oxygen (1 l/min) and 5% isoflurane and thereafter reduced to oxygen (1 l/min)/2% isoflurane for continuation. Rats were intubated with a 14G polyethylene tube and volume-controlled ventilated (Harvard apparatus model 683) according to the following settings: tidal volume of 6.5 ml/kg of body weight (BW), Positive End-Expiratory Pressure (PEEP) of 3 cmH$_2$O and an inspiration:expiration ratio of 1:1. Fraction of inspired oxygen (FiO$_2$) was 1, and after completion of BD induction reduced to 0.5. Respiratory frequency was initially 120/min and titrated throughout the experiment to keep end-tidal CO$_2$ (ETCO$_2$) between 20–22 mmHg.

The left femoral artery and vein were cannulated for mean arterial pressure (MAP) monitoring and fluid administration. In case of blood pressure drops below 80 mmHg, colloidal solution (HAES-steril 100 g/l, Fresenius, Bad Homburg, Germany) was administered with a maximum volume of 1 ml/h. In case of unresponsiveness to HAES, noradrenalin (NA, 0.01 mg/ml) was administered per infusion.

In rats subjected to BD, a fronto-lateral hole was drilled through the skull (Dremel, Breda, Netherlands), after application of lidocaine drops (10 mg/ml, Pfizer, Capelle aan den IJssel, the Netherlands) for local analgesia. Thereafter, a Fogarty catheter (Edwards Lifesciences LLC, Irvine, U.S.A.) was inserted in the epidural space. The total time needed for preparation steps from induction of anesthesia to insertion of the Fogarty catheter was 30 min. In the fast BD model, the catheter was expanded with 0.41 ± 0.02 ml saline over 1 min with a syringe perfusor pump (Terufusion Syringe Pump, model STC-521). In the slow BD model, the catheter was expanded with 0.41 ± 0.03 ml saline over 30 min. Rocuronium bromide (0.6 mg/kg of BW, Fresenius) was administered to reduce muscle movements during BD induction and body temperature was stabilized at 38°C with a heating pad. Upon completion of BD induction, anesthesia was withdrawn. The total duration of anesthesia induction to withdrawal was 31 min for the fast BD group and 60 min for the slow BD group. BD was confirmed by absence of corneal reflexes at 30 min and 1 hour after BD induction was completed. An additional apnea test was performed in the pilot experiment to validate the BD model, in absence of anesthesia and rocuronium. However due to fast desaturation, this test was omitted in the experiment.

Prior to organ harvest, suxamethonium chloride (0.1 mg/kg of BW, Fresenius) was administered to prevent spinal reflex movements. A laparo-thoracotomy was performed and blood was collected from the aorta, after which the circulatory system was flushed with 40 ml cold saline. Lungs were procured after inflation with 2 ml air. Control rats were euthanized by exsanguination upon direct procurement of organs, under anesthesia with a mixture of oxygen (1 l/min) and 5% isoflurane. Since control rats were not intubated, lungs were not inflated upon procurement.

## RNA isolation and RT-qPCR

RT-qPCR analyses were performed to detect pro-inflammatory gene expression levels in lungs. Total RNA was isolated from snap-frozen lung tissue with Trizol (Invitrogen Life Technologies, Breda, Netherlands) according to manufacturer's instructions, and RNA integrity was analyzed by gel electrophoresis. cDNA synthesis was performed according to manufacturer's instructions. Primer sets (Table 1) were loaded with 5 μl cDNA (2ng/μl) and SYBR green (Applied Biosystems). Amplification and detection were performed with the Taqman Applied Biosystems 7900-HT RT-qPCR system (Biosystems, Carlsbad, USA), measuring SYBR green emission. PCR reaction consisted of 40 cycles at 95°C for 15 s and 60°C for 60 s after initiation for 2 min at 50°C and 10 min at 95°C. Dissociation curve analyses ensured amplification of specific products. Gene expressions were corrected for appropriate housekeeping genes (β-actin, EIF2b1 and PPIA) and calculated with the ΔΔCt method [24].

**Table 1. RT-qPCR primers.**

| Primer | Gene | Forward Primer | Reverse Primer | Amplicon (bp) |
|---|---|---|---|---|
| Tnf-α | Tumor necrosis factor-alpha | AGGCTGTCGCTACATCACTGAA | TGACCCGTAGGGCGATTACA | 67 |
| Il-6 | Interleukin-6 | CCAACTTCCAATGCTCTCCTAATG | TTCAAGTGCTTTCAAGAGTTGGAT | 89 |
| Cinc-1 | Chemokine (C-x-C motif) ligand-1 | TGGTTCAGAAGATTGTCCAAAAGA | ACGCCATCGGTGCAATCTA | 78 |
| Ccl-2 (Mcp-1) | Chemokine (C-C motif) ligand-2 | CTTTGAATGTGAACTTGACCCATAA | ACAGAAGTGCTTGAGGTGGTTGT | 78 |
| Vcam-1 | Vascular adhesion molecule-1 | TGTGGAAGTGTGCCCGAAA | ACGAGCCATTAACAGACTTTAGCA | 84 |
| C3 | Central complement component 3 | CAGCCTGAATGAACGACTAGACA | TCAAAATCATCCGACAGCTCTATC | 96 |
| B-actin | Beta-actin | GGAAATCGTGCGTGACATTAAA | GCGGCAGTGGCCATCTC | 74 |
| Eif2b1 | Eukaryotic translation initiation factor 2B | ACCTGTATGCCAAGGGCTCATT | TGGGACCAGGCTTCAGATGT | 77 |
| Ppia | Peptidylprolyl isomerase A | TCTCCGACTGTGGACAACTCTAATT | CTGAGCTACAGAAGGAATGGTTTGA | 76 |

## Plasma analysis

The BD-induced catecholamine storm and inotropic support might substantially affect the heart and subsequently the lung [25]. To investigate heart injury, plasma levels of troponin and creatine kinase-MB (CK-MB) were determined in the clinical laboratory.

## Histological lung injury

Formalin-fixed, paraffin embedded lung slices (4 μm) were scored for histological injury after hematoxylin-eosin staining, according to a semi-quantitative scoring system as previously described [22]. Briefly, lungs were scored for: A) intra- and extra-alveolar hemorrhage, B) intra-alveolar edema, C) inflammatory infiltration of the inter-alveolar septa and airspace, D) over-inflation and E) erythrocyte accumulation below the pleura. Variables A-D were graded as: 0 = negative, 1 = slight, 2 = moderate, 3 = high, and 4 = severe. Variable E was scored as 0 = absent or 1 = present. Lungs were scored by two blinded investigators, with a conventional light microscope at a magnification of 200x across 10 random, non-coincidental fields.

## Immunohistochemistry

The number of activated neutrophils in lung tissue was quantified after myeloperoxidase (MPO) staining of paraffin embedded lungs. After deparaffinization, antigen retrieval was performed and sections were blocked for 30 min with endogenous peroxidase. Primary polyclonal rabbit anti-human antibody MPO (dilution 1/500, cat. No. A0398, Dako, Carpenteria, CA, USA) was incubated for 1 hour at room temperature. Thereafter, secondary antibodies (horse-radish peroxidase (HRP)-conjugated goat anti-rabbit, dilution 1/100, cat. No. P0448, Dako) and tertiary antibodies (HRP-conjugated rabbit anti-goat, dilution 1/100, cat. No. P0160, Dako) were incubated for 30 min. Reaction was developed through addition of 3,3'-diamino-benzidine-peroxidase substrate solution and sections were counterstained with hematoxylin. ImageJ Software (National Institutes of Health, Bethesda, MD, USA) was used to quantify 50 snapshots per lung on a 400x magnification.

## Statistical analysis

Statistical analyses were performed with IBM SPSS 22 (IBM corporation, New York, USA). The effect of BD model and time was examined by two-way mixed ANOVA tests for physiological parameters (pulmonary airway pressure (Paw), heart rate (HR), gene expressions, histological scoring, total volume administration, NA, CK-MB and troponin). Outliers and normality distributions were assessed by boxplot and probability-probability plot inspections. Not normally distributed data were transformed by the natural logarithm, after 0.1 was added to correct zero values of total volume administration and histological scoring. In case of non-linear changes over time (MAP, MCP-1 and VCAM-1), polynomials were applied prior to statistical tests. To determine differences between BD models, Mann-Whitney tests were performed (MAP, NA, HR, CK-MB, troponin, edema, Paw and neutrophil infiltration). Associations between BD model and pleura infarction were assessed by a Chi-square test, followed by Phi and Cramer's V for the strength of association. Data from rats with failure of hemodynamic stabilization were excluded from analysis. P<0.05 was considered statistically significant. Data are presented as mean ± SD, unless mentioned otherwise.

## Results

### Fast BD induction negatively affects hemodynamic stability of the donor

To investigate whether the mode of BD affects hemodynamic stability, we compared MAP, HR, required inotropic support and total fluid administration between the two BD models. In the 20 min before end of BD induction, MAP significantly decreased in the slow BD model (Fig 2). At the initial phase of hemodynamic stabilization (t = 0), MAP returned to pre-intervention values in the slow BD model. In contrast, peak MAP levels of rats subjected to fast BD did not return to pre-intervention values at t = 0. After 4 hours of stabilization, MAP levels were comparable between groups, although need of total fluid administration was higher in the fast BD model (1.2 ± 1.1 ml *versus* 0.6 ± 0.5 ml, p = 0.003). Thereby, more inotropic support with noradrenalin was required (0.72 ±1.07 ml *versus* 0.11 ± 0.26 ml) and HR was elevated in the fast BD model (Fig 3A–3B). CK-MB and troponin were assessed as heart injury markers, since the catecholamine storm and inotropic support might detrimentally affect the heart and subsequently the lung [25]. CK-MB release in plasma was higher in the fast BD model than in the slow BD model at time point 0.5 hours, although differences between groups were comparable after 4 hours of BD. CK-MB levels were not affected by time (Fig 4A). Troponin levels increased over time in both BD models, but did not differ significantly between fast and slow BD induction (Fig 4B). Collectively, these results suggest that rats subjected to fast BD induction are more hemodynamically compromised and require more assertive donor stabilization measures than rats subjected to slow BD induction.

### Fast BD induction results in more severe histological lung injury

Subsequently, we assessed whether severity of histological lung injury is affected by the mode of BD induction. In the fast BD group, six rats were lost due fulminant lung edema immediately after BD induction. This diagnosis was presented by visible lung fluid in the ventilation tube,

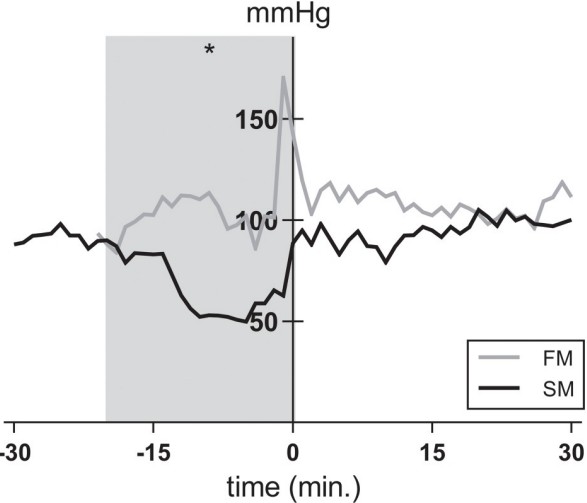

**Fig 2. Mean arterial pressure.** Rats were subjected to fast brain death (BD) induction (intracranial pressure (ICP) increase over 1 min) *versus* slow BD induction (ICP increase over 30 min). Expansion of the Fogarty catheter and induction of BD was finished at time point zero in both models. After BD was induced, all animals were stabilized above a mean arterial pressure of 80 mmHg for 0.5–4 hours, of which the first 30 min are presented in this figure. Values are depicted as mean. Data displayed in the grey area are significant between groups, as indicated by the asterisk: * p<0.05. FM–fast BD induction model, expansion of the Fogarty catheter over 1 min; SM–slow BD induction model, expansion of the Fogarty catheter over 30 min.

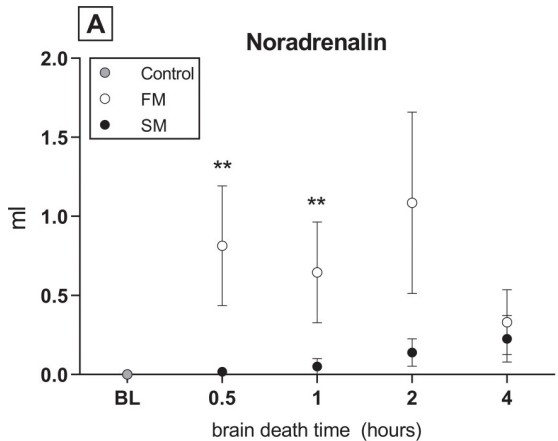
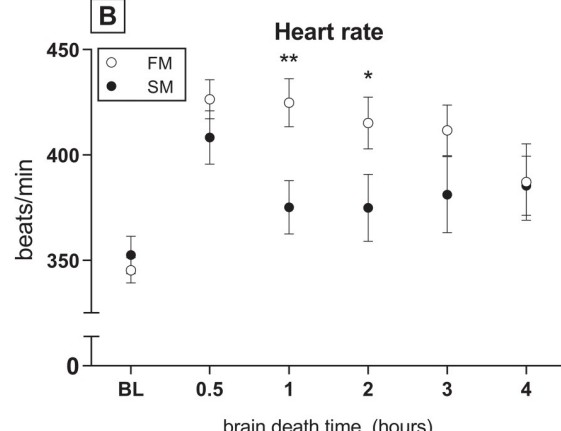

**Fig 3. Inotropic support and heart rate.** Rats were subjected to fast brain death (BD) induction (intracranial pressure (ICP) increase over 1 min) *versus* slow BD induction (ICP increase over 30 min) and subsequently stabilized for 0.5–4 hours on a mean arterial pressure of > 80 mmHg. (A) Required noradrenalin administration for hemodynamic stabilization. (B) Heart rate of brain-dead rats during the BD stabilization period. Values are presented as mean ± SEM. Asterisks denote significant differences between the two BD models per time point: * p<0.05, ** p<0.01. X-axis 0.5–4hours–brain-dead group with period of ventilation and hemodynamic stabilization time; BL–baseline measurement before BD induction; Control–immediately sacrificed without intervention; FM–fast BD induction model, expansion of the Fogarty catheter over 1 min; SM–slow BD induction model, expansion of the Fogarty catheter over 30 min.

with a subsequent inability to ventilate and macroscopic appearance of lung edema at dissection of the thoracic cavity. In contrast, no mortality occurred in the slow BD model. Histological lung injury scores increased over time and were significantly higher in the fast BD model (5.09 ± 2.11 at 4 hours after fast BD induction *versus* 4.38 ± 1.24 at 4 hours after slow BD induction, p = 0.040, Fig 5). This was the result of more pronounced hemorrhagic lung parenchyma in the fast BD model (1.70 ± 1.43 at 4 hours in the slow model *versus* 0.26 ± 0.33 at 4 hours in the fast model, p = 0.010) and more evident lung edema (0.81 ± 1.19 at 4 hours in the fast model *versus* 0.013 ± 0.04 at 4 hours in the slow model, p = 0.035). Furthermore, we observed a strong

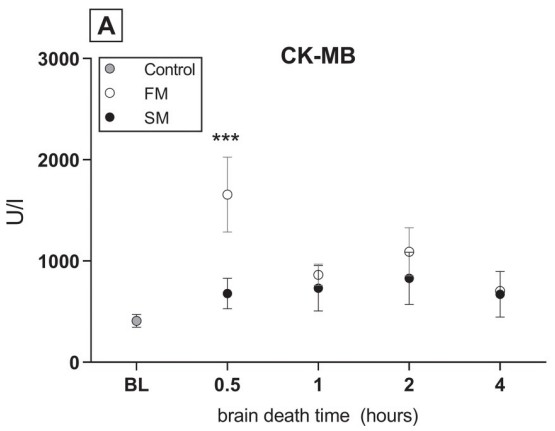
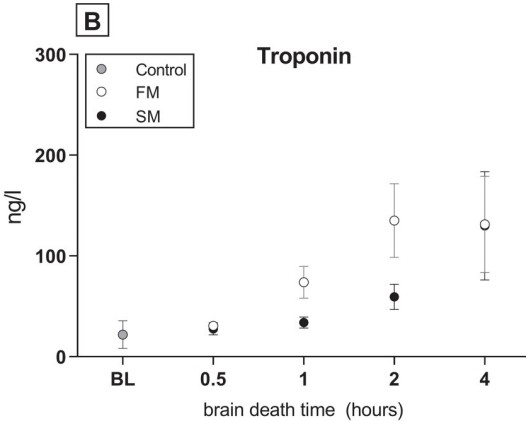

**Fig 4. Heart muscle injury markers CK-MB and troponin.** Rats were subjected to fast brain death (BD) induction (intracranial pressure (ICP) increase over 1 min) *versus* slow BD induction (ICP increase over 30 min) and subsequently stabilized for 0.5–4 hours on a mean arterial pressure of > 80 mmHg. (A) Plasma CK-MB levels and (B) plasma troponin levels of brain-dead rats and controls. Values are presented as mean ± SEM. Asterisks denote significant differences between the two BD models per time point: *** p<0.001. X-axis 0.5–4 hours–brain-dead group with period of ventilation and hemodynamic stabilization time; BL–baseline measurement before BD induction; CK-MB–Creatine kinase (myocardium); Control–immediately sacrificed without intervention; FM–fast BD induction model, expansion of the Fogarty catheter over 1 min; SM–slow BD induction model, expansion of the Fogarty catheter over 30 min.

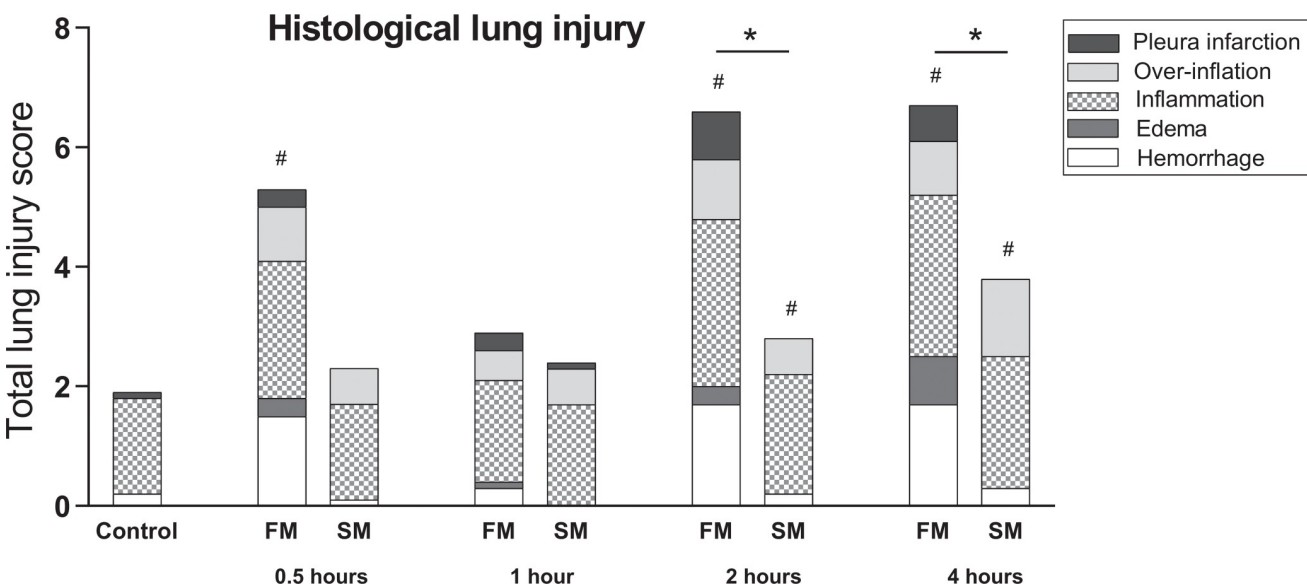

**Fig 5. Histological lung injury score.** Rats were subjected to fast brain death (BD) induction (intracranial pressure (ICP) increase over 1 min) *versus* slow BD induction (ICP increase over 30 min). Lungs were procured after 0.5–4 hours of donor stabilization and histological lung injury scores were assessed. All values are presented as mean. Asterisks denote significant differences between the two BD models per time point: * p<0.05. '#'-symbols indicate significant differences compared to controls. X-axis 0.5–4 hours–brain-dead group with period of ventilation and hemodynamic stabilization time; Control–immediately sacrificed without intervention; FM—fast BD induction model, expansion of the Fogarty catheter over 1 min; SM—slow BD induction model, expansion of the Fogarty catheter over 30 min.

association between the fast BD induction model and pleura infarction (p = 0.000). Despite results on histological lung injury, no differences were observed in Paw between the BD models (Fig 6). Taken together, these results indicate that histological lung injury is more pronounced after fast induction of BD when compared to a slow induction of BD.

## Fast and slow BD induction lead to a comparable pulmonary immune response

Pulmonary inflammation after fast *versus* slow BD-induction was investigated through analyses for pro-inflammatory gene expressions and infiltration of activated neutrophils. In both BD models, gene expression levels of TNF-α, IL-6, IL-8-like CINC-1, MCP-1, VCAM-1 and central complement component C3 were significantly increased compared to controls (Fig 7A–7F). For MCP-1 and VCAM-1 an increase in a quadratic regression was noted over time, with time centered at 1.5 hours. Between the BD models, expressions of TNF-α, MCP-1 and VCAM-1 showed significant differences at time point 0.5 hours. TNF-α and MCP-1 expression levels were higher in the fast BD model, while VCAM-1 was more pronounced in the slow BD model. However, from 1 hour after BD induction and after, cytokine levels were comparable between the two BD models. Additionally, infiltration of activated neutrophils after 4 hours did not differ between the models, as depicted by the number of MPO-stained leukocytes (Fig 8A–8D). Collectively, these results suggest that the BD-induced immune response after donor stabilization is comparable between fast and slow BD induction.

## Discussion

The cause of BD substantially influences early outcomes after transplantation, as described for kidney and heart transplantation [26, 27]. However, regarding donor lungs, studies failed to

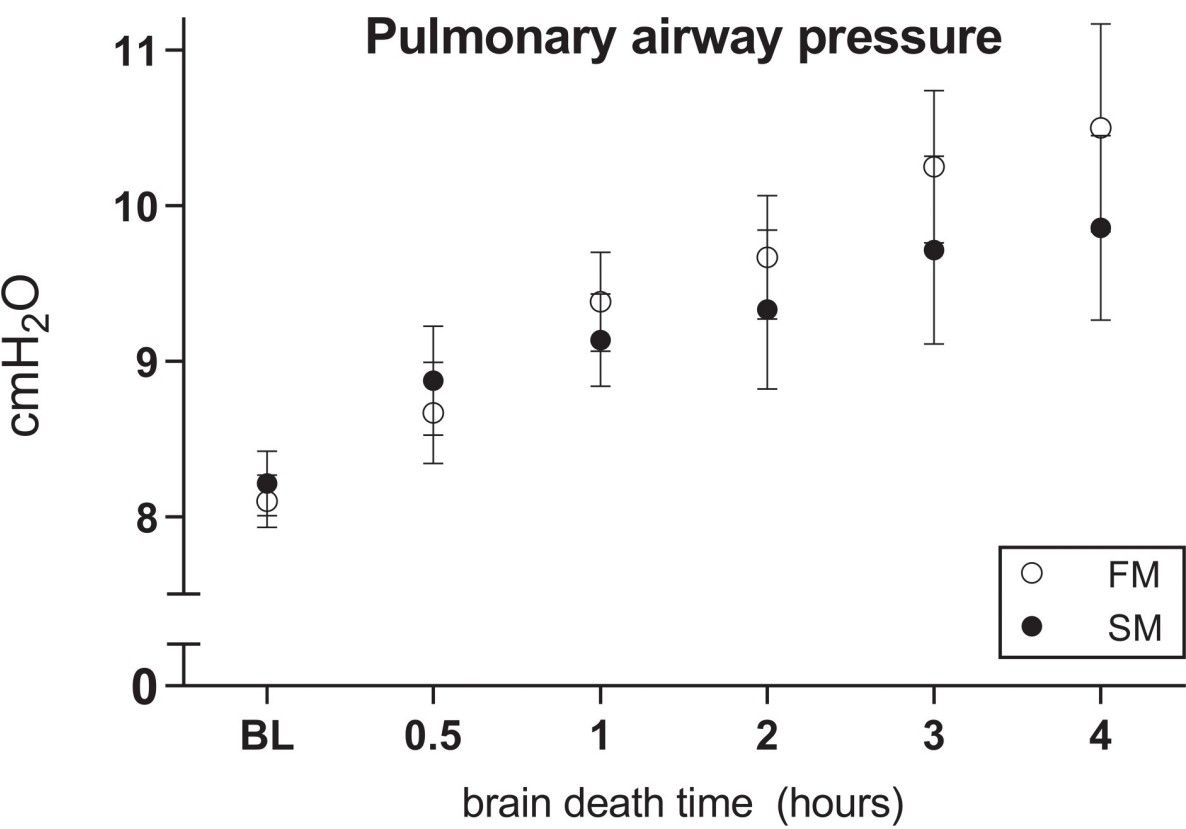

**Fig 6. Pulmonary airway pressure.** Rats were subjected to fast brain death (BD) induction (intracranial pressure (ICP) increase over 1 min) *versus* slow BD induction (ICP increase over 30 min). Subsequently, rats were hemodynamically stabilized and ventilated for 0.5–4 hours and pulmonary airway pressures were noted. Values are presented as mean ± SEM. X-axis 0.5–4hours–brain-dead group with period of ventilation and hemodynamic stabilization time; BL–baseline measurement before BD induction; FM–fast BD induction model, expansion of the Fogarty catheter over 1 min; SM–slow BD induction model, expansion of the Fogarty catheter over 30 min.

show a correlation between cause of BD and graft outcome in terms of survival [28–30]. Whether the cause of BD contributed to a decreased quality and availability of the donor lung, was not investigated in these studies. The aim of our study was to elucidate whether the speed of ICP increase affects quality of lungs from brain-dead donors, investigated in a rat model for fast *versus* slow BD induction. This study showed that fast BD induction deteriorates donor lung quality more than slow BD induction.

First, we studied the effect of fast *versus* slow BD induction on hemodynamic stability of rats. We observed higher MAP during the initial phase of the hemodynamic stabilization period in the fast BD model, and more fluid administration and inotropic support was required. These results support the observation that rats subjected to fast BD induction show deteriorated hemodynamic stability when compared to rats subjected to slow BD induction, as previously described in a comparative study with emphasis on the heart [19]. In the mentioned study, the acute MAP increase in the fast BD model was described as the result of more pronounced sympathetic discharge, followed by more profound hypotension. In accordance, we found a more pronounced elevation of heart enzymes in the fast BD model than in the slow BD model, which probably reflects a more extensive intrinsic catecholamine release [19].

Next, we investigated whether the mode of BD affects the degree of histological lung damage. The observation that more rats were lost due to fulminant lung edema directly after BD induction in the fast BD model, together with higher histological lung injury scores, strongly suggest that

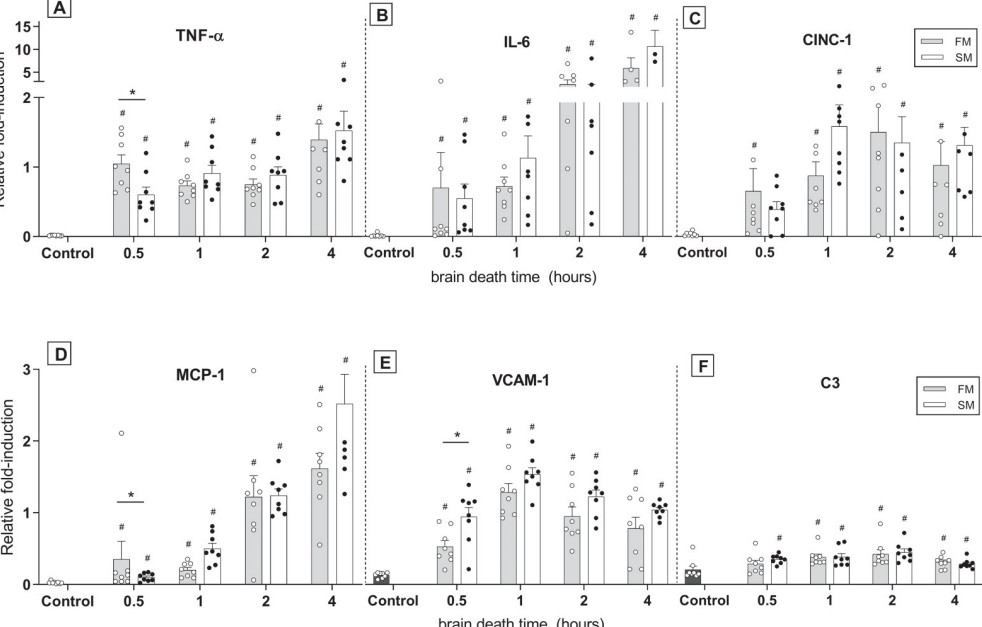

**Fig 7. Pro-inflammatory gene expressions.** Rats were subjected to fast brain death (BD) induction (intracranial pressure (ICP) increase over 1 min) *versus* slow BD induction (ICP increase over 30 min). Lungs were procured after 0.5–4 hours of donor stabilization and the following gene expression levels for pro-inflammatory cytokines were assessed by means of RT-qPCR: (A) TNF-α, (B) IL-6, (C) IL-8-like CINC-1, (D) MCP-1, (E) VCAM-1 and (F) C3. Data are shown as expression relative to housekeeping genes PPIA, ELf2b1 and β-actin. Values are presented as mean ± SEM. Asterisks denote significant differences between the two BD models per time point: * p<0.05. '#'-symbols indicate significant differences compared to controls. X-axis 0.5–4hours–brain-dead group with period of ventilation and hemodynamic stabilization time; Control–immediately sacrificed without intervention; FM–fast BD induction model, expansion of the Fogarty catheter over 1 min; SM–slow BD induction model, expansion of the Fogarty catheter over 30 min.

lung damage is more evident after fast BD induction. Increased histological lung injury scores were the result of more pronounced hemorrhagic infarcted lung parenchyma and edema. This is possibly due to the observed differences in MAP, since a sudden MAP increase is described to rupture the capillary-alveolar membrane and disrupt barrier integrity [15]. Thereby, Avlonitis *et al.* demonstrated that changes in capillary-alveolar membrane integrity are prevented when the hypertensive response is eliminated [15]. These observations could explain why acute cerebral insults and BD have been associated with the onset of pulmonary edema [11, 31, 32], while this is rarely observed in subarachnoid hemorrhage [33], an example of gradual ICP increase.

Last, we investigated whether differences in lung damage were accompanied by a different inflammatory status in fast *versus* slow BD models. Despite that the more pronounced histological lung damage upon fast BD was evident up to 4 hours after BD induction, inflammatory status was comparable between BD models at 4 hours of BD. Nevertheless, at time point 0.5 hours, we found higher TNF-α and MCP-1 levels after fast BD induction than after slow BD induction. These results might be explained by the observed course in MAP, since a correlation is suggested between hemodynamic changes and the inflammatory immune response during BD [15, 34, 35]. Indeed, higher MAP in the fast BD model preceded increased TNF-α and MCP-1 levels in the fast BD model. Thereby, from 1 hour of BD and after, inflammatory status as well as MAP were comparable between groups. Another possible explanation for comparable gene expressions after the initial BD period is development of prerenal acute kidney injury (AKI) in the slow BD model, due to the observed hypotensive phase [36]. The

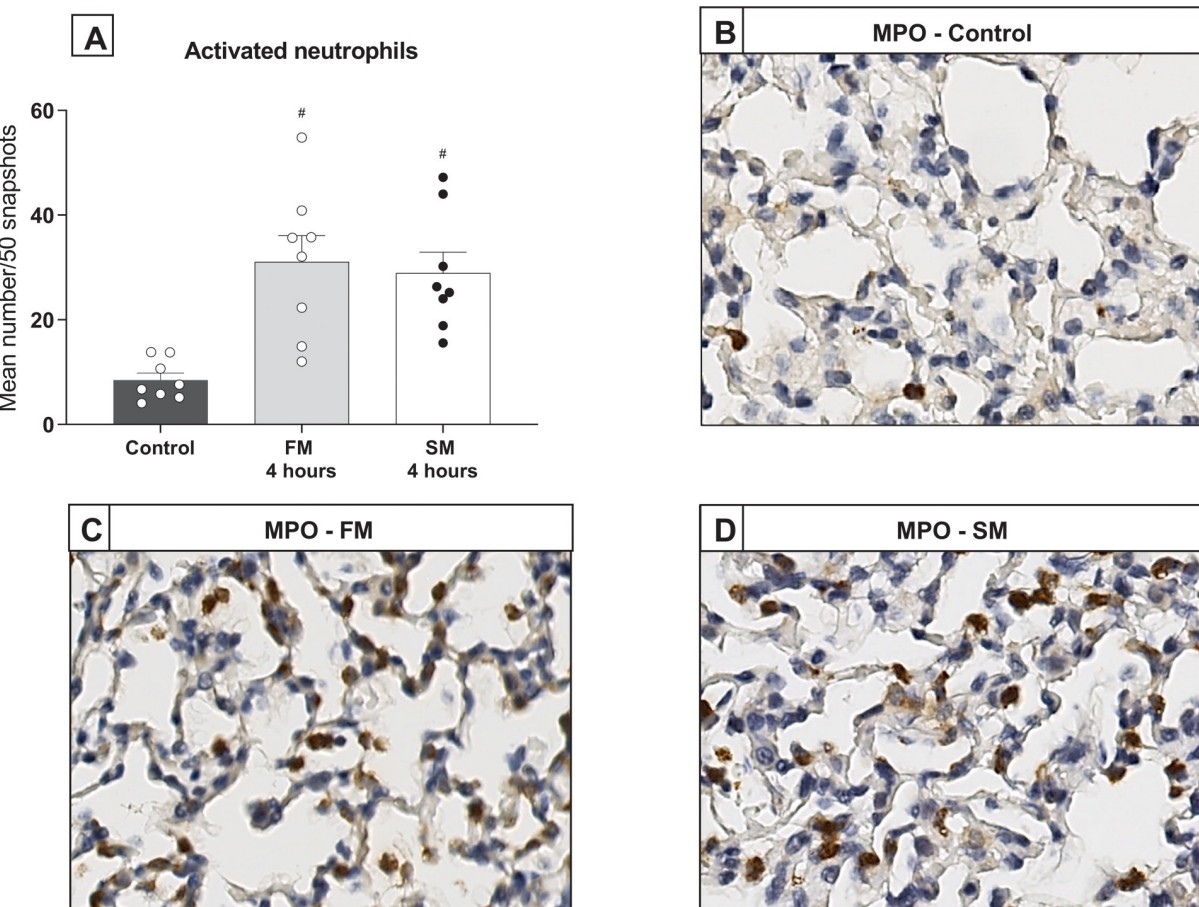

**Fig 8. Infiltration of activated neutrophils.** Rats were subjected to fast brain death (BD) induction (intracranial pressure (ICP) increase over 1 min) *versus* slow BD induction (ICP increase over 30 min). Lungs procured after 4 hours of donor stabilization were stained for myeloperoxidase (MPO) to assess infiltration of activated neutrophils. (A) Quantification of neutrophils as depicted by MPO staining. (B-D) Representative MPO-stained lung slides of controls and rats subjected to fast or slow BD induction. Values are presented as mean ± SEM. '#'-symbols indicate significant differences compared to controls. Control–immediately sacrificed without intervention; FM—fast BD induction model, expansion of the Fogarty catheter over 1 min; SM—slow BD induction model, expansion of the Fogarty catheter over 30 min.

double hit of ischemic AKI and onset of BD might have increased pro-inflammatory gene expressions in the slow BD model, which possibly resulted in a similar degree of pulmonary inflammation as determined in the fast BD model [37, 38].

Previous experimental studies have extensively investigated the pathophysiological mechanisms of BD in controlled models [5]. However to our knowledge, a comparison of fast *versus* slow induction of BD and the effect on lung quality has not been described before. Despite that experimental models provide the opportunity to study BD-related pathophysiology in a controlled manner, it should be noted that the heterogeneity of BD in humans makes a direct translation to the clinical setting inaccurate. Previously performed clinical studies investigated the relation between mode of BD and transplantation outcomes, however with inconsistent results. An early retrospective study of Waller *et al.* compared outcome of donors involved in major trauma to donors with nontraumatic origin, and showed no differences in early complications after lung transplantation [28]. However in a later retrospective study by Ciccone *et al.*, traumatic BD seemed to predispose to higher rejection episodes in the first year after lung transplant and subsequent development of bronchiolitis obliterans syndrome, though the

mechanisms involved were not identified [39]. Pilarczyk *et al.* found no differences in early complications or 1 and 3-year survival in their prospective study comparing traumatic and non-traumatic causes of BD, yet 5-year survival was suggested to be lower in recipients from donor lungs subjected to traumatic BD [40]. However, it should be noted that none of these clinical studies reported whether the mode of BD affected donor lung suitability. Therefore, injured donor lungs due to traumatic causes of BD might therefore have been excluded. In addition, the time-point of evaluation of potential donor lungs is extremely variable between donors, which contributes to the heterogeneity of BD-induced injury in clinical donors.

In this study, we focused on the isolated effect of two modes of BD on donor lung quality in homogenous groups, in contrast to the clinical setting. The fast and slow BD model are well established in our laboratory, which we both adjusted to enable a direct comparison between models [21, 22]. The fixed time-point of lung procurement at 4 hours after induction of BD contributed hereto. Since the primary outcome measure in our study was quality of donor lungs at time of retrieval, we chose to not include a transplant model in the current study. Nevertheless, we consider our controlled rat study to be an important contribution to the current, fundamental knowledge on BD-induced lung injury. Possible clinical implications might include a broader approach to the selection of potential donor lungs, especially in patients who suffered from traumatic causes of BD. Thereby, understanding the pathophysiological mechanisms in different modes of BD may inspire to a more customized approach in donor management to improve donor lung quality. Albeit not directly compared to slow induction modes of BD, earlier studies already suggested that protective lung ventilation strategies seem particularly important in the setting of acute, massive brain damage [41]. Furthermore, the application of *ex-vivo* perfusion strategies may facilitate the tailored approach to improve donor lung quality, when other potential donor organs may benefit from different optimization strategies. As our group previously published, abdominal organs seem to suffer more from detrimental consequences of slow BD induction than fast BD induction [36]. Therefore, *ex-vivo* treatment may further improve lung quality in an isolated manner and thereby increase the pool of suitable donor lungs.

## Conclusion

In conclusion, this study accentuates the consequence of BD mode on donor lung quality, by demonstrating that fast BD induction deteriorates quality of donor lungs more on a histological level than slow BD induction, while inflammatory levels were comparable.

## Supporting information

**S1 Dataset. Dataset of Figs 2–8.**
(PZFX)

**S1 Fig.**
(TIFF)

**S1 File.**
(PDF)

**S1 Checklist. The ARRIVE guidelines 2.0: Author checklist.**
(PDF)

## Acknowledgments

We would like to thank J. Zwaagstra, P. Ottens and S. Veldhuis for their technical support throughout the project.

## Author Contributions

**Conceptualization:** Rolando A. Rebolledo, Dane Hoeksma, Jeske M. Bubberman, Annette Breedijk, Benito A. Yard, Michiel E. Erasmus, Henri G. D. Leuvenink, Maximilia C. Hottenrott.

**Formal analysis:** Judith E. van Zanden, Johannes G. Burgerhof.

**Methodology:** Judith E. van Zanden, Rolando A. Rebolledo, Dane Hoeksma, Jeske M. Bubberman, Johannes G. Burgerhof, Annette Breedijk, Maximilia C. Hottenrott.

**Supervision:** Benito A. Yard, Michiel E. Erasmus, Henri G. D. Leuvenink, Maximilia C. Hottenrott.

**Writing – original draft:** Judith E. van Zanden.

**Writing – review & editing:** Judith E. van Zanden, Rolando A. Rebolledo, Dane Hoeksma, Jeske M. Bubberman, Johannes G. Burgerhof, Annette Breedijk, Benito A. Yard, Michiel E. Erasmus, Henri G. D. Leuvenink, Maximilia C. Hottenrott.

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
