## [Decision Letter · Decision Letter 0]

7 Sep 2020

PONE-D-20-21408

Rat donor lung quality deteriorates more after fast than slow brain death induction

PLOS ONE

Dear Dr. van Zanden,

Thank you for submitting your manuscript to PLOS ONE. After careful consideration, we feel that it has merit but does not fully meet PLOS ONE’s publication criteria as it currently stands. Therefore, we invite you to submit a revised version of the manuscript that addresses the points raised during the review process.

ACADEMIC EDITOR:

Interesting and well-executed study, reviewed by 3 experts in the field, who have come up with major queries to be addressed in detail by the authors regarding the clinical relevance and/or translation of the findings, and reviewer 2 particularly challenges some of the important conclusions drawn by the authors. I invite the authors to make the revisions required in a detailed manner, and answer the concerns of the reviewers one by one.

We look forward to receiving your revised manuscript.

Kind regards,

Frank JMF Dor, M.D., Ph.D., FEBS, FRCS

Academic Editor

PLOS ONE

Journal Requirements:

Reviewers' comments:

Reviewer's Responses to Questions

**Comments to the Author**

1. Is the manuscript technically sound, and do the data support the conclusions?

Reviewer #1: Yes

Reviewer #2: Partly

Reviewer #3: Yes

2. Has the statistical analysis been performed appropriately and rigorously? 

Reviewer #1: Yes

Reviewer #2: Yes

Reviewer #3: Yes

3. Have the authors made all data underlying the findings in their manuscript fully available?

Reviewer #1: Yes

Reviewer #2: Yes

Reviewer #3: Yes

4. Is the manuscript presented in an intelligible fashion and written in standard English?

Reviewer #1: Yes

Reviewer #2: Yes

Reviewer #3: Yes

5. Review Comments to the Author

Reviewer #1: This paper is part of a long history of research on the effects of donor brain death on outcome after organ transplantation by the group of Leuvenink et al. In this study, they show in rats that slowly induced brain death evokes less trauma and inflammation to the lungs than rapidly induced brain death. Although this suggests that outcomes after lung transplantation are better when using donors who suffer from intracranial hemmorhage (slow) compared with traumatic brain injury (fast), several clinical studies show similar outcomes between both groups. In the present study, the authors refrained from transplanting lungs from both groups, and therefore were not able to assess in their model the effect of the different brain death modalities.

The authors conclude by stating that "in future research, our aim is to focus on pre-conditioning strategies of brain death donors in order to limit lung inflammation and improve transplantation outcome". Since clinical data suggest lungs from fast brain death donors do not perform worse, it is questionable that this will improve outcome. Although the results are interesting in itself, and the study is well executed, my main question therefore is how this study may contribute to improving the outcome after lung transplantation.

Some minor comments:

Line 142: please change bleeding out in exanguination.

Line 160: why were cardiac injury parameters measured. And then why not also renal injury markers?

Line 240 and other figures: For clarity please add x-axis before 0.5-4 h.

Reviewer #2: The evaluation of the influence of the speed of brain death occurrence on the quality of the donor organs is an important finding for the improvement of the donor management approach. In this regard, the study of van Zanden and co-workers investigated the donor lung quality in a rat model of fast versus slow BD induction. Despite to be a well-designed protocol, the authors need to clarify several aspects before to have the manuscript approved for publication. Furthermore, this reviewer strongly disagreed with the conclusion that this study demonstrates that fast BD induction deteriorates quality of donor lungs more than slow BD induction, as after donor stabilization, inflammatory status was comparable between the two BD models.

Regarding the hemodynamic stability, after an expected initial instability in the fast induction model, MAP levels were comparable between groups after 4 hours of stabilization. Moreover, CK-MB and Troponin release in plasma were also comparable between groups after 4 hours of BD. The authors described the use of more inotropic support with noradrenalin in the fast BD model, but it is not clear if this fact could really influence the late stabilization of the hemodynamic behavior observed in both groups.

The description that fast BD induction resulted in more severe histological lung injury was only based on the fact that six rats were lost due to inability of ventilation in that group and on a combined histological lung injury score. Regarding the rats lost due to ventilation problems, the authors need to show histological data to confirm the proposed diagnosis of fulminant lung edema, as others authors have published similar studies with the fast induction model without deaths for up to 6 hours of follow-up. It was described that histological lung injury scores were significantly higher in the fast BD model because of more pronounced hemorrhagic lung parenchyma and lung edema. However, it would be very important to know the statistical differences observed in each one of these alterations to really define the level of the lung parenchyma deterioration, as Infiltration of activated neutrophils and pro-inflammatory gene expressions were similar in both groups.

Reviewer #3: This is an elegant and well performed piece of work from a group who have made significant contributions to clinical pulmonary transplantation.

There is a good introduction, summarising much of the literature. However, they give little emphasis or even acknowledgement that the work done is almost all in experimental models of brain death, with a very standardised injury. Thus, the patterns seen are well described in the experimental setting. Whilst there are some reports correlating the cause of donor death and donor lung function, the huge heterogeneity of brain death in the clinical setting of the organ donor must be acknowledged and discussed

A subset of donors, particularly if they have suffered traumatic brain injury, have undergone ICP monitoring. It is from these donors that the initial hypotheses about rapidity of the whole process and organ damage. In the experimental model, the rate of inflation of the intracranial balloon varied. Was intracranial pressure, and the way it changed, examined?

Whilst the observations made are novel and appropriate markers of both haemodynamic and inflammatory damage have been recorded, there is little or no discussion about how this may translate into improved clinical care. This again highlights the gap between rigid models of brain death in the rat and the heterogeneity of the phenomenon in the human organ donor. Rather than just making a series of correct but rather sterile observations, the authors should relate their findings to clinical management.

There is an uncited paper about speed of brain death and renal injury:

Kerforne T, Giraud S, Danion J, et al. Rapid or Slow Time to Brain Death? Impact on Kidney Graft Injuries in an Allotransplantation Porcine Model. Int J Mol Sci. 2019;20(15):3671. Published 2019 Jul 26. doi:10.3390/ijms2015367

6. PLOS authors have the option to publish the peer review history of their article (what does this mean?). If published, this will include your full peer review and any attached files.

Reviewer #1: No

Reviewer #2: No

Reviewer #3: **Yes: **John Dark

---

## [Author Response · Author response to Decision Letter 0]

20 Oct 2020

Dear Dr. Dor,

Thank you for providing us the opportunity to submit a revised version of our manuscript ‘Rat donor lung quality deteriorates more after fast than slow brain death induction’ to PLOS ONE. We are grateful for the time that you and the reviewers dedicated to provide us feedback on our manuscript. We have incorporated most of the comments and suggestions of the reviewers, which are highlighted in yellow in the revised manuscript. In our attached ‘response to reviewers’ file, you find our point-by-point response to the reviewers’ concerns and suggestions. 

Yours sincerely,

J.E. van Zanden

Dear reviewer #1,

We thank you for your constructive comments and valuable suggestions for our manuscript. In this reply, we address the issues raised point-by-point, as best as possible.

- The cause of donor death in relation to transplantation outcome has indeed been investigated in previous studies. An early retrospective study of Waller et al. (Waller et al. JHLT, 1995) already compared outcome of donors involved in major trauma to donors with nontraumatic origin, and showed that mode of donor death does not influence early complications after lung transplantation (Waller et al. JHLT, 1995). In a later study, Ciccone et al. suggested in their retrospective study that early results after lung transplantation might not be affected by donor cause of brain death. However, traumatic brain injury seemed to predispose to higher rejection episodes in the first year after transplantation and subsequent development of bronchiolitis obliterans syndrome, though the mechanisms involved were not identified (Ciccone et al. J. Thorac. Cardiovasc. Surg. 2002). Since the retrospective nature of these studies may be accompanied with significant bias, prospective studies are of additional value. In the study of Pilarczyk et al. data was prospectively collected, in which they compared traumatic cause of donor death to non-traumatic cause of donor death and the effect on outcome after lung transplantation. No differences were found in early complication or 1 and 3-year survival, though 5-year survival was suggested to be lower in recipients from donor lungs subjected to a traumatic cause of death (Pilarczyk et al. Thorac. Cardiovasc. Surg. 2017). 

However, an important issue that was not addressed in these clinical studies, is whether the mode of donor death affected donor lung suitability. For example, lungs from instable patients in the ICU might not be offered for potential organ donation. As Pilarczyk also discussed in their manuscript, otherwise injured donors due to traumatic brain injury may therefore be excluded, leading to exclusion bias (Pilarczyk et al. Thorac. Cardiovasc. Surg. 2017). In addition, the extensive heterogeneity in the clinical setting should be noted, which complicates a direct comparison between different modes of brain death. Amongst others the time-point of evaluation of a potential donor lung is variable between donors, partly due to logistical factors. For these reasons, we consider a controlled experimental study of additional value to the current knowledge of mode of brain death and donor lung quality. In our experimental rat study all donor lungs were evaluated at a fixed time point of 4 hours after brain death induction. We included all potential donor rats in the results of our manuscript, thereby no exclusion bias was present in our study. 

Furthermore, we would like to emphasize that our group previously published the results of fast versus slow brain death with emphasis on the abdominal organs. The mentioned study showed that abdominal organs, in contrast to the lungs, seem to suffer more from detrimental consequences of slow brain death induction than fast brain death induction (Rebolledo et al. J. Transl. Med. 2016). The different observations between lungs and abdominal organs inspire to a more customized approach in donor management to improve donor lung quality. Possibly, the application of ex-vivo perfusion strategies may play herein an important role.

We elaborated on these thoughts in the ‘discussion’ section of our manuscript, highlighted in yellow.

Minor comments

- Line 142: we changed ‘bleeding out’ to ‘exsanguination’ (changes highlighted in yellow). 

- Line 160: cardiac injury markers were assessed, since the brain death-induced catecholamine storm and the administered inotropic support during brain death might affect the heart, and subsequently the lung (mentioned in the ‘methods’ section of the manuscript). Renal injury markers were previously assessed and published by our group, in which fast versus slow brain death was investigated with emphasis on kidney and liver function (Rebolledo et al. J. Transl. Med. 2016). In the mentioned study, our group showed that slow brain death induction is more detrimental for potential donor kidneys, as corroborated by higher plasma creatinine and urea levels. In addition, IL-6 gene expression levels were higher in kidneys subjected to slow brain death induction than in kidneys subjected to fast brain death induction. Since we published these results before we did not include them in the current manuscript, though we highlighted these previous findings in the ‘discussion’ section (changes highlighted in yellow).

- Line 240 and other figures: ‘X-axis’ was added before 0.5-4 h for clarity of figure legends (changes highlighted in yellow).

Once again, we would like to thank you for the time you put in reviewing our paper. Hopefully, we were able to meet your expectations and we would like to thank you in advance for your reply.

Dear reviewer #2,

We thank you for your constructive comments and valuable suggestions for our manuscript. In this reply, we address the issues raised point-by-point, as best as possible.

We are challenged by the comment that the reviewer strongly disagrees with the conclusion of our study, and we are very willing to clarify our view on this matter to contribute to this intriguing discussion and implemented some changes. With regard to hemodynamic stability, MAP levels were indeed comparable between the two modes of brain death after 4 hours of stabilization. However, we would like to emphasize that this stabilization was the result of efficient donor management with administration of colloids and noradrenalin. Our group has perfected our established brain death model over the years, and from experience we learned that without efforts to maintain MAP > 80 mmHg, many donor rats are lost. For that reason, we chose to stabilize MAP > 80 mmHg with administration of colloids and noradrenalin. Figure 2 therefore represents signs of efficient donor management, in which MAP was stabilized to comparable values between the two brain death groups, after the initial MAP instability in the fast brain death model. However, as we described in the ‘results’ section of our manuscript, rats subjected to fast brain death induction required indeed more fluid administration and inotropic support than rats subjected to slow brain death. These results suggest that true MAP of rats subjected to fast brain death would have been lower if MAP was not stabilized, with the possible risk to lose the animal. We believe that the more pronounced elevation of CK-MB and reflects a more extensive intrinsic catecholamine release in the fast brain death model, as we wrote in the ‘discussion’ section of our manuscript. In addition, we consider the stabilization of heart enzymes after 4 hours of brain death as the result of efficient donor management and MAP stabilization. 

Indeed after 4 hours of brain death, total lung injury scores were worse in lungs from rats subjected to fast brain death induction than in lungs subjected to slow brain death induction. This was the result of amongst others more evident lung edema, which might be amplified by the need of more inotropic support and fluid administration in the fast brain death model. To clarify the differences for these significant variables between the two modes of brain death, we added the specific values with corresponding p-values to the ‘results’ section of our manuscript. In the fast brain death group, 6 rats were indeed lost due to fulminant lung edema within few minutes after brain death induction. Unfortunately, we did not collect samples from these donor lungs and are therefore unable to show histological data. However macroscopically, this diagnose was presented through presence of fulminant lung edema in the ventilation tube and subsequent inability to ventilate the donor rats. Technical problems with the ventilator were excluded. Therefore, we are confident that fulminant lung edema caused the loss of these particular rats subjected to fast brain death. We clarified this observation in the ‘results’ section of the manuscript. 

Indeed, previously published studies by the group from Mannheim under supervision of Prof. Dr. B. Yard describe successful fast brain death induction sustained for 6 hours follow-up. For example in the paper by Krebs et al. (Crit. Care 2014), different ventilation strategies were examined. We would like to emphasize that the last author of our study (M. Hottenrott) took part in the experiments of the mentioned manuscript, and brought the fast brain death model to Groningen. In the Groningen laboratory, the slow brain death model was already established, which enabled us to perform the direct comparison after adaptation of both models. From previous experiments, we observed that rats subjected to slow brain death induction became instable after 4 hours of brain death induction. Therefore, we terminated the investigative period after 4 hours. However, in the current study, we optimized hemodynamic management before brain death induction and introduced a standardized and less deleterious lung ventilation. With these protocol improvements we observed that when rats survived the first minutes after brain death induction, they were of no risk to be lost during follow-up. Therefore, we are confident that with the current protocol, both models could be continued for 6 hours. 

Taken together, we believe that based on these results, fast brain death induction deteriorates donor lung quality more than slow brain death induction, in particular on a histological level. Nevertheless, we agree that the similar inflammatory status between groups is an important observation as well, and therefore we modified our concluding paragraph in the ‘abstract’ and ‘conclusion’ section of our manuscript.

Once again, we would like to thank you for the time you put in reviewing our paper. Hopefully, we were able to satisfy you with our adaptations and we would like to thank you in advance for your reply.

Dear reviewer #3,

We thank you for your constructive comments and valuable suggestions for our manuscript. In this reply, we address the issues raised point-by-point, as best as possible.

- We agree with the reviewer that the tremendous heterogeneity of brain death in the clinical setting is distinctly different from standardized, experimental models. Therefore, direct translation to the clinical setting is inaccurate. We elaborated on this matter in a new paragraph of the ‘discussion’ section of our manuscript (changes highlighted in yellow). 

- In our model, the difference between fast versus slow brain death induction was indeed accomplished by inflation of the balloon catheter over different time rates. We did not measure intracranial pressure in our donor rats, though we appreciate this suggestion. However, in previous attempts to measure intracranial pressure in our model, no reliable results were obtained. This might be due to the small animal size and the therefore corresponding challenge to insert an ICP catheter in the cerebral ventricle.

- We agree with the reviewer that speculations about clinical implications of our results will add more depth to our paper. For that reasons, we added an extra paragraph to the ‘discussion’ section, in which we addressed this issue. 

Once again, we would like to thank you for the time you put in reviewing our paper. Hopefully, we were able to meet your expectations and we would like to thank you in advance for your reply.

---

## [Decision Letter · Decision Letter 1]

10 Nov 2020

Rat donor lung quality deteriorates more after fast than slow brain death induction

PONE-D-20-21408R1

Dear Dr. van Zanden,

We’re pleased to inform you that your manuscript has been judged scientifically suitable for publication and will be formally accepted for publication once it meets all outstanding technical requirements.

Kind regards,

Frank JMF Dor, M.D., Ph.D., FEBS, FRCS

Academic Editor

PLOS ONE

Additional Editor Comments (optional):

Reviewers' comments:

Reviewer's Responses to Questions

**Comments to the Author**

1. If the authors have adequately addressed your comments raised in a previous round of review and you feel that this manuscript is now acceptable for publication, you may indicate that here to bypass the “Comments to the Author” section, enter your conflict of interest statement in the “Confidential to Editor” section, and submit your "Accept" recommendation.

Reviewer #1: (No Response)

Reviewer #2: All comments have been addressed

2. Is the manuscript technically sound, and do the data support the conclusions?

Reviewer #1: (No Response)

Reviewer #2: Yes

3. Has the statistical analysis been performed appropriately and rigorously? 

Reviewer #1: (No Response)

Reviewer #2: Yes

4. Have the authors made all data underlying the findings in their manuscript fully available?

Reviewer #1: (No Response)

Reviewer #2: Yes

5. Is the manuscript presented in an intelligible fashion and written in standard English?

Reviewer #1: (No Response)

Reviewer #2: Yes

6. Review Comments to the Author

Reviewer #1: (No Response)

Reviewer #2: The authors addressed all the questions raised and changed satisfactory the comments and conclusions based on the limitations of their study.

7. PLOS authors have the option to publish the peer review history of their article (what does this mean?). If published, this will include your full peer review and any attached files.

Reviewer #1: No

Reviewer #2: No

---

## [Editor Report · Acceptance letter]

16 Nov 2020

PONE-D-20-21408R1 

Rat donor lung quality deteriorates more after fast than slow brain death induction 

Dear Dr. van Zanden:

I'm pleased to inform you that your manuscript has been deemed suitable for publication in PLOS ONE. Congratulations! Your manuscript is now with our production department. 

Kind regards, 

on behalf of

Dr. Frank JMF Dor 

Academic Editor

PLOS ONE